# Integrated Fluidic Platform for Washing and Mechanical Processing of Lipoaspirate for Downstream Fat Grafting and Regenerative Applications

**DOI:** 10.3390/bioengineering12090918

**Published:** 2025-08-26

**Authors:** David Zalazar, Jiayi Feng, Derek A. Banyard, Marzieh Aliaghaei, Alan D. Widgerow, Jered B. Haun

**Affiliations:** 1Department of Biomedical Engineering, University of California Irvine, Irvine, CA 92697, USA; dzalazar@uci.edu (D.Z.); jfeng8@uci.edu (J.F.); 2Sayenza Biosciences Inc., Irvine, CA 92617, USA; derek@sayenza.com; 3Department of Chemical and Biomolecular Engineering, University of California Irvine, Irvine, CA 92697, USA; m.aliaghaie@gmail.com; 4Department of Plastic Surgery, School of Medicine, University of California Irvine, Orange, CA 92868, USA; awidgero@hs.uci.edu; 5Center for Tissue Engineering, University of California Irvine, Orange, CA 92868, USA; 6Department of Materials Science and Engineering, University of California Irvine, Irvine, CA 92697, USA; 7Chao Family Comprehensive Cancer Center, University of California Irvine, Orange, CA 92868, USA; 8Center for Advanced Design and Manufacturing of Integrated Microfluidics, University of California Irvine, Irvine, CA 92697, USA

**Keywords:** fat grafting, closed-loop system, mechanical processing, nanofat, stromal vascular fraction, tissue engineering

## Abstract

Autologous fat grafting of human lipoaspirate (LA) is increasingly used in reconstructive and cosmetic surgery for lipofilling and stem cell-rich “nanofat” reinjection for regenerative medicine. While commercial devices (e.g., REVOLVE and Puregraft) are available, many surgeons use non-standardized manual washing techniques, leading to inconsistent graft retention (20–80%). Moreover, no system can unite washing directly with mechanical processing to produce a nanofat-like product directly from raw LA. We developed a novel preparation device (PD) that is designed for peristaltic pump-driven washing of LA and can be seamlessly combined with our previously developed Emulsification and Micronization Device (EMD) into an automated closed-loop platform. Human LA samples were washed with the PD and compared to standard manual washing via visual colorimetric analysis. We then evaluated the mechanical processing of PD-washed LA using our EMD and assessed cell count, viability, and stromal vascular fraction-derived subpopulations (i.e., mesenchymal stem cells, endothelial progenitor cells (EPCs), pericytes, transit-amplifying (TA) progenitor cells, and supra-adventitial adipose stromal cells). Recirculating LA through the PD for at least one minute resulted in sufficient mixing, producing LA with equivalent color and quality to manual washing. Integrating the EMD within a platform enabled both washing and mechanical processing under peristaltic flow, enriching key subpopulations compared to manual methods. Thus, our fluidic platform effectively washes LA in a closed-loop system, minimizing LA tissue manipulation and opportunity for contamination while also simplifying the workflow for mechanical processing. Further refinement and automation of this platform would enhance the reproducibility and quality of small-volume fat grafts, cell-assisted lipotransfer, and stem/progenitor cell injections to promote wound healing and angiogenesis.

## 1. Introduction

Human lipoaspirate (LA) is commonly used for autologous fat grafting in plastic- and reconstructive surgeries and is increasingly being utilized for regenerative medicine. As the largest source of stem cells in the body [1,2,3], adipose-derived therapeutics are being explored across a wide range of conditions, ranging from osteoarthritis [4,5,6,7,8] to diabetic foot ulcers [9,10,11,12,13] to cardiac repair [14,15,16,17]. Over the years, a wide range of regenerative and pro-angiogenic cell types have been located within the adipose-derived stromal vascular fraction (SVF), including endothelial cells, endothelial progenitor cells (EPCs), mesenchymal stem cells (MSCs), pericytes, and macrophages [18,19]. Furthermore, Zimmerlin et al. proposed that a CD34+ subset of CD146+ pericytes was present within SVF, functioning as a transit-amplifying (TA) progenitor cell capable of giving rise to both EPCs and supra-adventitial adipose stromal cells (SA-ASCs) [20]. This population has demonstrated regenerative benefits when supplemented to fat grafts, increasing retention and vascularization, improving histological quality, and reducing resorption in pre-clinical mouse models [21,22]. Altogether, SVF is an appealing catalyst for wound healing, promoting angiogenesis, and microvascular remodeling in vivo [23,24,25,26].

In 2013, Tonnard et al. introduced nanofat (NF), an enzyme-free mechanical processing method to emulsify adipose tissue by shuffling washed LA between two syringes for 30 passes [27]. The advent of this technique enabled a concentrated source of stem and progenitor cells derived from the SVF to become readily accessible to surgeons with limited regulatory hurdles [28,29]. Although commercial devices (i.e., Tulip [30,31], Lipogems [32,33], Lipocube Nano [34,35], etc.) are on the market, they result in high variability given that manual shuffling is still required. We recently developed the Emulsification and Micronization Device (EMD) [36], which applies enhanced and more consistent hydrodynamic shear forces using a syringe pump to mechanically break down LA aggregates. Compared to traditional NF processing, our EMD enriched MSC and EPC populations [36] and displayed enhanced wound healing transcriptional programs and angiogenesis in vitro [37]. While the enhanced NF produced by our EMD holds therapeutic promise, the need to manually wash LA limits processing workflow speed and versatility, particularly in clinical settings.

Washing LA is a critical step for both graft fillers and SVF applications, but it is non-standardized, and this can contribute to variable outcomes. Saline is typically added at an equal volume to LA, followed by mixing, gravity separation, and decantation of the aqueous infranatant containing contaminants such as blood, debris, and tumescent solution [38,39]. This process is repeated at least three times, or until the desired visual appearance is obtained. While decantation is the most common technique [40,41,42,43,44], centrifugation [41,43,44,45] is another popular option. In recent years, commercialized devices such as Viality [46,47], Puregraft [48,49,50], Revolve [49,51], and Genesis Biosystems have improved processing workflow [52,53,54], yet fat graft retention rates still remain low [50,55]. Moreover, these systems only perform wash procedures, leaving mechanical processing to another device or operation. A platform that could perform both washing and mechanical processing in a seamless manner would dramatically advance indications utilizing either autologous fat as graft fillers or SVF as an injectable therapeutic. This would be particularly exciting for combining the two products, known as cell-assisted lipotransfer (CAL), which has been shown to improve graft retention. The evidence has been strong in clinical settings using culture-expanded ADSCs [56,57], but the results using SVF-enriched grafts are less clear. While small-volume facial grafts have demonstrated benefits [58], results with larger-volume grafts to the breasts have demonstrated mixed results [59,60]. Langridge et al. (2023) hypothesized that the lack of clinical significance in larger-volume grafts is likely due to insufficient ASC enrichment [55]. Meanwhile, animal models lend support to the idea that SVF-enrichment of grafts must be optimized to achieve improved outcomes [61,62]. Thus, there is a critical need for standardization of LA washing and SVF generation, ideally with a single integrated platform, to help decrease user-induced variability from sample preparation and enable automation.

Here, we present a novel closed-loop LA preparation platform that enables both washing and mechanical processing within the same system (Figure 1). The preparation device (PD) is a novel technology that washes LA with saline by mixing and agitating via recirculating peristaltic flow and internal spiral baffles. Using human LA, we first optimize the PD recirculation time to produce comparable color and clarity to manual washing. Next, we mechanically process PD-washed LA using our EMD with a syringe pump and find similar enrichment of stem and progenitor cell subtypes to manual washing. We observed that longer recirculation times through the PD did result in lower enrichment levels, suggesting that overprocessing can occur. Finally, we demonstrate a fully integrated closed-loop workflow in which the PD was used both for washing and as a holding reservoir during EMD processing with the peristaltic pump. By uniting washing and mechanical processing devices into a single platform, we can achieve greater consistency and efficacy for all regenerative applications utilizing SVF. We also envision utility for small-volume fillers and reconstructive surgeries utilizing SVF for CAL.

## 2. Materials and Methods

### 2.1. Human Subjects and Ethical Considerations

All recruited patients were actively undergoing liposuction for either cosmetic or reconstructive procedures. Active systemic infection or use of immunosuppressive therapy resulted in exclusion from participation. A total of 13 patients were recruited for this study. One patient’s demographic and procedural data were missing from the records, but their inclusion was accounted for in the total patient count. The cohort included 11 females and 1 male, with an average age of 49 years (range 29 to 63 years). The racial and ethnic composition was as follows: 7 Caucasian patients (58%; 6 females, 1 male), 2 Hispanic patients (17%), 1 Black/African American patient (8%), 1 Persian patient (8%), and 1 Indian patient (8%). Adipose specimens were collected from the back (17%), flanks (25%), thighs (17%), abdomen (33%), and chest (8%) using standard vacuum-assisted liposuction with 3- (13%), 3.7- (13%), 4- (7%), 4.6- (13%), and 5 mm (53%) harvest cannulas and stored at room temperature until use. No additional exclusions occurred based on missing data.

### 2.2. Preparation Device and Emulsification and Micronization Device

The EMD was fabricated as previously described [36]. Both the EMD and PD were 3D-printed as a single piece by Dinsmore Inc. (Irvine, CA, USA) using an SLA 3D printer and biocompatible Somos BioClear resin from Royal DSM (Elgin, IL, USA). The PD contains a cylindrical body with spiral baffles extending along the entire inner body to increase the mixing between LA and saline (Figure 2A). These spiral baffles are fin-shaped and angled downward with a height of 3.1 mm, a top width of 1.6 mm, and base width of 5.7 mm. The pitch between adjacent spiral ribs is 21 mm. The PD contains two inlets, one for processing and another for venting, both located on a threaded cap. The PD also contains an outlet at the distal portion of the cylindrical body that allows for bi-directional LA flow. Both inlets contain ¼ barbed connectors, and the outlet contains threading for Luer-Lok connections with an ID of 3.9 mm. The volume of the PD is ~100 mL; however, once tubing and valves are attached, the total volume increases to ~150 mL. A photograph of the fabricated PD is shown in Figure 2B.

For operation of the EMD with a syringe pump, a Harvard Apparatus PhD Ultra Syringe Pump (Harvard Apparatus, Holliston, MA, USA) was utilized at a flow rate of 20 mL/s. For peristaltic LA washing and EMD processing, a Masterflex L/S Digital Drive Peristaltic Pump was used with an attached Masterflex L/S Easy Load II CRS Pump Head and Masterflex L/S size 24 and size 35 tubing, all obtained from VWR (Radnor, PA, USA). Three-way valves were obtained from Burkle (Item: 8608-0080, Bad Bellingen, Germany) and were utilized to switch between flow paths. Peristaltic pump flow rates ranged from 5 mL/s (for LA washing) to 20 mL/s (for LA EMD processing).

### 2.3. PD Device Operation

The PD was mounted on a laboratory stand and affixed with tubing, three-way valves, and Luer-to-Barb connectors (Figure 3). To initiate loading of the PD, valves were configured to block the distal outlet of the PD and enable access from the saline inlet through both the processing loop and the LA inlet. The venting inlet was opened to load without pressure buildup within the PD. Loading was performed with 50 mL BD Luer-Lok tip syringes (Item: 309653, Franklin Lakes, NJ, USA) through both a male Barb to Luer adaptor (Item: 01-000-156, Fischer Scientific, Waltham, MA, USA) and female-to-female Luer adaptor (Item: LPP-FF, Component Supply, Sparta, TN, USA) inserted within the tubing at the saline inlet. This Barb-to-Luer configuration was present at both the saline and LA inlets, as well as the waste outlet to allow for syringe loading/unloading of PD contents.

Approximately 75 mL of saline was loaded through the saline inlet to preload the processing loop. This quantity is sufficient to fill the processing loop and approximately a fourth of the inner wash cavity of the PD. After saline loading, ~75 mL LA can be loaded into the inner wash cavity through the LA inlet until the device is full. After loading, the valves were configured to close the vent and open the processing loop to the PD. The peristaltic pump was driven clockwise at a rate of 5 mL/s for the specified time interval (see Appendix A), and samples were allowed to gravity-separate for 5 min between each washing round.

After gravity separation concluded, the infranatant waste fraction was removed from the PD and tubing lines. First, the vent was opened and a 50 mL syringe was connected to the waste outlet connector prior to positioning the waste valve to allow for flow from the PD. The resultant infranatant layer was removed. Next, the waste outlet valve was positioned to allow access to the processing loop and the contents were removed until LA began to enter the syringe. Following waste removal, the appropriate volume of saline was added to the system through the saline inlet by resetting the valve configuration to the starting positions. Once loaded, the workflow outlined above was repeated to complete three washes. Finally, after the waste infranatant was removed, LA was collected through the saline inlet valve using a 50 mL syringe for downstream applications. For PD-integrated mechanical processing, valves were configured to allow access to the EMD (Figure 1), and both clockwise direction and flow rate of 20 mL/s were used for the specified processing time interval (see Appendix A).

### 2.4. Cell Analysis

Washed human LA was digested with collagenase and analyzed as previously described [36]. Briefly, LA was mixed at a 1:1 volume ratio mixture with 0.1% type I collagenase (Sigma-Aldrich Co., St. Louis, MO, USA) to release the SVF. The mixture was incubated at 37 °C for 30 min while swirling every 5 min, and digestion was quenched through addition of control media (DMEM supplemented with 10% fetal bovine serum, 500 IU penicillin, and 500 μg streptomycin). The mixture was allowed to gravity-separate for 10 min, and the infranatant layer containing SVF was collected. Recovered infranatant was then filtered through a 100 μm cell strainer (Fischer Scientific, Waltham, MA, USA), centrifuged at 500× *g* for 7 min, and resuspended in PBS for cell counting. An automated dual-fluorescence cell counter and AO/PI Viability Dye (Logos Biosystems Inc., Annandale, VA, USA) were used to count nucleated cells and determine viability.

Cell subtypes (Table 1) were identified by flow cytometry, as previously reported [36]. Briefly, isolated cells underwent staining with monoclonal antibodies (Table 2, all from BioLegend, San Diego, CA, USA) for 20 min at 4 °C in PBS containing 1% BSA (PBS+). Cells were then resuspended in PBS+ supplemented with 7-AAD (BD Biosciences, San Jose, CA, USA) so as to exclude dead cells and analyzed on a Novocyte 3000 Flow Cytometer (ACEA Biosciences, San Diego, CA, USA). Appropriate single stains were performed with both cells and Invitrogen UltraComp eBeads (Fischer Scientific, Waltham, MA, USA) for generating a compensation matrix. Heat-kill and Fluorescence Minus One (FMO) controls were used to determine signal positivity of all antibodies and viability stains (Appendix A). Flow cytometry data analysis was performed using FlowJo version 10.10.0 software (Ashland, OR, USA). Our analysis included two pericyte gating strategies (Appendix A) given separate marker sets in the literature—pericytes [20,63,64,65] and CD34- pericytes [66,67,68].

### 2.5. Statistics

Data are presented as mean ± standard error based on a minimum of three independent patient samples. Normality of residuals was assessed using both the D’Agostino–Pearson omnibus K^2^ test and the Shapiro–Wilk test. Homogeneity of variance was evaluated using Spearman’s test for heteroscedasticity. When assumptions of normality and equal variance were met, a randomized block two-way ANOVA was conducted [69], followed by Holm–Šídák’s multiple comparisons post hoc test. For data that violated ANOVA assumptions, non-parametric comparisons were performed using the Friedman test with Dunn’s post hoc test. Statistical significance was defined as *p* < 0.05. All analyses were performed using GraphPad Prism version 10.4.2 (GraphPad Software, Boston, MA, USA).

## 3. Results

### 3.1. Optimization of PD Washing Performance

We first assessed the effect of recirculating peristaltic flow through the PD on mixing of human LA (n = 6) and saline, as well as ultimate wash efficacy. The collected infranatant fractions from the 1-min recirculation time intervals were similar in color and clarity to manually washed fractions (Figure 4A). Recirculating LA samples longer through the PD resulted in visually darker eluent fractions. The final collected LA samples did not result in observable color changes between the PD and manually washed samples (Figure 4B). Additionally, all the infranatant fractions and LA colors were substantially less dark than the unwashed LA sample, confirming successful washing (Figure 4A,B). The cellular content of the washed LA was also assessed after collagenase digestion using an automated cell counter. The nucleated cell counts were highest for the 1-min recirculation condition at 770,000 ± 125,000 cells/mL LA (Figure 4C). The 3-min recirculation condition had a cell count of 690,000 ± 120,000 cells/mL LA, whereas both MF and the 5-min recirculation condition had lower cell counts at 630,000 ± 120,000 and 640,000 ± 93,000 cells/mL LA, respectively, although the differences were not statistically significant. All the wash conditions resulted in cellular viability between 88 and 90%, and the differences were not significant (Figure 4D).

We further evaluated PD-washed LA by processing with the EMD operated with a syringe pump (sEMD), as in previous work [36,37]. sEMD processing resulted in lower but comparable cell counts across most conditions: MF 550,000 ± 94,000 cells/mL LA, PD (1 min) 570,000 ± 81,000 cells/mL LA, and PD (3 min) 540,000 ± 64,000 cells/mL LA. Conversely, the PD (5 min) condition had a higher cell count at 650,000 ± 66,000 cells/mL LA. Both the PD wash- and syringe pump-processed conditions resulted in cellular viability >87% (Figure 4D). Nucleated cell count and viability were not statistically different for any conditions.

Finally, we evaluated key SVF cell subpopulations for all the wash conditions, with and without sEMD processing, using flow cytometry. Subpopulations included macrophages, MSCs, EPCs, pericytes, and TA progenitor cells (Figure 4E–L). All the values were reported as % total cells to show enrichment (Appendix A) and then normalized to MF to exhibit differences from this key control condition. Interestingly, PD washing alone resulted in modest enrichment of all the cell types over MF, in the range of 20–50%, suggesting there may be some level of mechanical processing involved while recirculating through the PD system. The lone exception was macrophages, which remained similar to MF. Following sEMD processing, we observed substantial enrichment of all the cell subpopulations relative to MF. However, manual and PD washing did not affect the ultimate enrichment levels, suggesting that mechanical processing from PD washing was not additive. The overall enrichment was approximately 2-fold for macrophages (Figure 4I) and MSCs (Figure 4E) and 3-fold for EPCs (Figure 4F). Modest enrichment was observed for both standard and CD34- pericyte populations (Figure 4G,H), at approximately 50%, and, notably, sEMD processing did not augment enrichment further than the PD-washing effect already discussed. Both the CD34^dim^ and SA-ASC subsets contained equivalent enrichment profiles for all the conditions, with sEMD and PD (1 min) + sEMD exhibiting a 20% increase compared to MF (Figure 4J,K). Contrastingly, the CD34^dim^ TA progenitor cell subset demonstrated slightly higher enrichments of 70% and 80% for the sEMD and PD (1 min) + sEMD conditions, respectively (Figure 4L).

Upon comparison of PD-washing, the enrichment levels for all the subpopulations remained consistent, with a slight downward trend with longer time. Specifically, we observed lower enrichment levels for the 5-min PD recirculation time after sEMD processing for macrophages, MSCs, EPCs, and pericytes, which will preclude this condition from further study. A detailed summary of cell counts and population percentages is provided in Appendix A. Statistical comparisons for all the outcomes and testing conditions are provided in Appendix A. Notably, there was a significant 60% enrichment in pericytes among all the PD wash conditions (*p* = 0.0164) compared to MF (Figure 4G). Both the 1-min (*p* = 0.0146) and 3-min (*p* = 0.0468) recirculation times also resulted in ~80% CD34^dim^ TA progenitor cell enrichment. Macrophages increased by 90% with sEMD processing and 100% for both the PD (1 min) and (3 min) conditions, although only the 1-min condition proved statistically significant (*p* = 0.0322) relative to MF. Increases in MSCs were 2.1-fold for both the sEMD (*p* = 0.0005) and PD (1 min) + sEMD (*p* = 0.0037) conditions, and 1.9-fold for the PD (3 min) + sEMD (*p* = 0.0049) condition (Figure 4E). In contrast, the PD (5 min) + sEMD condition exhibited a significant decrease in MSCs compared to sEMD (*p* = 0.0218). The EPCs exhibited the highest levels of enrichment, with the PD (1 min) + sEMD condition increasing 2.9-fold (*p* = 0.0026), the PD (3 min) + sEMD condition increasing 2.5-fold (*p* = 0.0218), and the PD (5 min) + sEMD condition increasing 2.1-fold (*p* = 0.0468) (Figure 4F).

### 3.2. Peristaltic EMD Processing Is Equivalent to Syringe Pumping

To adapt our previous sEMD findings [36] towards a platform that can be operated entirely with peristaltic pumping, manually washed human LA (n = 4) was processed with the EMD at a flow rate of 20 mL/s using a peristaltic pump to recirculate flow (pEMD), as schematically shown in Figure 5A. Peristaltic processing was performed within a tubing loop for 30, 60, or 90 s. Assuming uniform flow through the EMD, these times correspond to 15, 30, and 45 passes using the sEMD protocol, respectively. The results were compared to sEMD using 30 passes, as used previously in this work and others. The total cell counts were similar for MF and all EMD processing (Figure 5B). Viability significantly differed across the conditions (ANOVA, *p* = 0.0107), with MF demonstrating higher viability than conventional sEMD (*p* = 0.0223), pEMD (30 s) (*p* = 0.0194), pEMD (60 s) (*p* = 0.0194), and pEMD (90 s) (*p* = 0.0157). Despite these differences, cell viability remained at >80% for all the conditions (Figure 5C). No significant differences in viability emerged between the sEMD and pEMD processing modalities.

Flow cytometry demonstrated that pEMD was able to match, if not surpass, the enrichment of sEMD for some subpopulations relative to MF (Figure 5D–K). Non-normalized cell subtype recovery values and statistical comparisons are provided in Appendix A. We observed a general trend of increasing enrichment with pEMD processing time, although the differences were small and not statistically significant. For MSCs (Figure 5D), pericytes (Figure 5F,G), macrophages (Figure 5H), and CD34^dim^ TA progenitors (Figure 5K), only 30 s of peristaltic flow was needed to match or surpass the sEMD results. However, EPCs required 60 s to match and 90 s to surpass sEMD (Figure 5E). A detailed summary of cell counts and population percentages is provided in Appendix A. Statistical comparisons for all the outcomes and testing conditions are provided in Appendix A. Specifically, EPCs demonstrated a significant 4.1-fold enrichment (*p* = 0.0118) with sEMD processing, as well as a non-significant 2.4-fold increase after 30 s of pEMD processing that further increased to a significant 3-fold increase after 60 s (*p* = 0.0422). After 90 s of pEMD processing, there was a significant 4.6-fold enrichment (*p* = 0.0033) compared to MF. We also note statistically significant differences after 90 s of pEMD processing for both pericytes (2.2-fold increase, *p* = 0.0438) and CD34- pericytes (60% increase, *p* = 0.0376) compared to MF.

### 3.3. Integrated Platform for PD Washing EMD Mechanical Processing

Given the successful mechanical processing of LA using a peristaltic pump, we integrated the EMD with the PD to create a single closed-loop platform (Figure 6A). The PD was employed both as the wash chamber and sample reservoir during EMD processing, which enabled the same peristaltic pump to be utilized for both tasks with flow pathways set via two valves. Given the new format for mechanical processing, we will refer to this system as the integrated EMD (iEMD). To validate the performance, we washed a human LA specimen (n = 3) with the PD using three rounds of saline addition and 1-min recirculation times and then mechanically processed with the sEMD or iEMD protocol. For the iEMD tests, we processed the entire volume of washed LA, which was ~4 times higher volume than previous pEMD studies. Thus, we scaled up the iEMD processing times accordingly to 180, 270, and 360 s (Figure 6B–K). While MF resulted in the highest cell counts at 400,000 ± 102,000 cells/mL, both sEMD and iEMD (180 s) conditions had comparable cell counts at ~380,000 cells/mL (Figure 6B). Increased processing time led to lower nucleated cell counts, dropping to 270,000 ± 17,000 cells/mL and 310,000 ± 107,000 cells/mL, respectively; however, these differences were not statistically significant. The viability for all the tested conditions was ≥90% except for MF at 84% (Figure 6C).

Following iEMD processing, we again observed dose-dependent enrichment for most cell types, which exceeded sEMD even at the lowest 180 s time point. The non-normalized cell subtype recovery values and statistical comparisons are provided in Appendix A. MSCs exhibited only a minimal 10% increase with sEMD processing but showed dose-dependent enrichment, with processing time ranging from 50% to 2.4-fold (Figure 6D). Similarly, EPCs displayed a comparable enrichment between the sEMD processing and iEMD (270 s) conditions at 80% and 90%, respectively (Figure 6E). At 360 s of iEMD processing, enrichment increased 2.8-fold but did not reach statistical significance. Both pericytes (ANOVA, *p* = 0.0162) and CD34- pericytes (ANOVA, *p* = 0.0019) exhibited significant differences in enrichment upon iEMD processing. For pericytes, iEMD enrichment was greater across all the processing times, with iEMD (180 s) displaying an 80% increase compared to sEMD. Notably, iEMD (360 s) exhibited a 4.4-fold increase compared to MF that was statistically significant (*p* = 0.0122) (Figure 6F). CD34- pericytes also had a statistically significant increase across all the iEMD processing conditions compared to MF (Figure 6G). iEMD processing at 180 s resulted in a 2.9-fold increase (*p* = 0.0307), followed by a 4.2-fold increase at 270 s (*p* = 0.0027). At 360 s, enrichment slightly declined to 3.9-fold increase (*p* = 0.0026). Both the 270- and 360-s iEMD conditions demonstrated significantly greater enrichment compared to sEMD processing (*p* = 0.0347). Macrophage enrichment increased by ~30% over MF at all the processing time points (Figure 6H). iEMD processing decreased the CD34^dim^ populations (Figure 6I) and CD34^dim^ SA-ASC (Figure 6J), with sEMD processing displaying an 80% increase and then decreasing to 40% and 30% after 180 s of iEMD processing, respectively. While the CD34^dim^ parent population decreased to 10–20% at longer iEMD durations, the CD34^dim^ SA-ASC subpopulation showed only a modest 5% decrease at 270 s and more pronounced 18% decrease at 360 s. In contrast, the CD34^dim^ TA progenitor subpopulation exhibited 3.1-fold enrichment following sEMD processing, which increased to 5.4-fold in the iEMD (180 s) condition and further to a statistically significant 12-fold increase at iEMD (360 s) compared to MF (*p* = 0.0246) (Figure 6K). A detailed summary of the cell counts and population percentages is provided in Appendix A. Statistical comparisons for all the outcomes and testing conditions are provided in Appendix A.

## 4. Discussion

Adipose tissue is used extensively for lipofilling, and adipose tissue-derived therapeutics hold exceptional promise in regenerative medicine, repurposing what was once considered surgical waste into a valuable therapeutic. This study introduces a novel device that standardizes and improves the workflow for washing LA. We demonstrated that a 1-min mixing algorithm using the PD can effectively wash LA without negatively affecting key cellular metrics (i.e., nucleated cell counts and viability) or stem/progenitor cell subpopulations. Furthermore, we demonstrated that PD-washed LA can be directly transitioned to mechanical processing within a single integrated system, thus being enriching for SVF subpopulations known to support healing and angiogenesis.

The commercial lipoaspirate preparation market can currently be divided into either washing or mechanical processing. Commercialized washing devices, like Puregraft and Revolve, utilize similar principles as the manual wash method employed in our study, including hand mixing and gravity separation. We would anticipate similar wash quality and time metrics to our new method using the PD. Commercial mechanical processing devices, like Tulip and Lipocube Nano, utilize manual shuffling of tissue similar to the nanofat method used in this study. We have already shown in published works that our EMD produces higher stem/progenitor cell enrichment and recovery, and this was confirmed with the new platform. Finally, the new peristaltic modality for mechanical processing simplified the system architecture and reduced processing time without compromising quality. Our integrated fluidic platform is the first, to our knowledge, that enables both washing and mechanical processing within one system. Designed with automation in mind, our platform is amenable to full automation, which will eliminate manual saline loading and waste removal steps and reduce both processing time and contamination risk.

We were foremost interested in understanding how the PD could be utilized to efficiently mix LA and saline via peristaltic pumping, aided by internal baffles. We hypothesized that longer wash intervals would result in cleaner LA given that tissue would mix as the sample recirculated through the wash chamber and was agitated by the baffles. When assessing the collected waste fractions and washed LA samples (Figure 4A,B), we observed that a 1-min recirculation time was comparable to MF. No observable differences were evident among the washed LA samples with increased wash times. A subsequent cellular analysis of the washed samples following collagenase digestion did not reveal significant changes among the nucleated cell counts (Figure 4C), viability (Figure 4D), or cell subpopulations of interest (Figure 4E–L). Conversely, mechanical processing of these washed samples with our EMD revealed a potential sensitivity of adipose tissue to prolonged wash intervals within the PD (Figure 4E–G,I–L). We hypothesize that increasing agitation of LA via peristaltic pumping and across baffles may mechanically process the sample to some degree, which was observed from PD-washed samples without EMD processing. Any mechanical processing within the PD will need to be carefully controlled so as not to overprocess the sample prior to use as a graft or SVF injection.

We also found that peristaltic EMD processing proved advantageous for enriching progenitor and perivascular populations (i.e., EPCs, pericytes, and CD34^dim^ TA progenitors). While we hypothesized that matching the theoretical number of passes between syringe and peristaltic pumping modalities would result in comparable results, we found that additional processing time was required on a per-sample volume basis, particularly for the iEMD case when the LA sample was scaled up. A possible explanation is that larger parcels of LA entering the EMD inlet may have had an impact on the flow rate for peristaltic processing. Peristaltic pumps do not produce high pressures, so any resistance within the flow path can cause back pressure that would limit the sample from passing through the EMD. Given the luminal nature of EPCs [70], it is also possible that these cells require longer durations of shear stress and/or greater applied pressure during processing for enrichment. Conversely, while we observed a slight decrease in pericyte enrichment with increased wash duration (Figure 4G), pericytes exhibited a statistically significant increase at longer processing durations (Figure 5F and Figure 6F). CD34- pericytes also appeared resistant to increased shear force (Figure 4H) and demonstrated significant improvements across all the pEMD durations (Figure 5G and Figure 6G). Pericytes have been shown to exist within a continuum of phenotypes [71] and demonstrate plasticity among the pericyte–vascular smooth muscle cell axis [72]. Interestingly, we noticed varying responses to shear stress between both pericyte populations (Figure 4, Figure 5 and Figure 6). It is unclear whether our phenotypic snapshot encompassed quick transcriptional changes as a response to shear stress [73], migrating pericytes from the capillary basement membrane [74], or non-capillary pericyte subsets [75]. Similarly, CD34^dim^ TA progenitors increased in a dose-dependent manner with peristaltic processing (Figure 5K and Figure 6K). As a transitional population between adventitial pericytes and supra-adventitial ASCs [20,70], it would be expected that these cells could remain stable in response to hydrodynamic pressure and shear stresses [76]. Interestingly, we observed that CD34^dim^ cells were sensitive to increased shear stress (Figure 4J, Figure 5I and Figure 6I). This smooth muscle cell subset has been shown to display limited proliferative and adipogenic capacity compared to the CD34+/CD34^bright^ subpopulation [77,78,79,80]. We found that SA-ASC subpopulations within the CD34^dim^ subset showed similar enrichment trends to the overall CD34^dim^ gate, decreasing with longer processing times (Figure 4J,K, Figure 5I,J and Figure 6I,J). These findings support our hypothesis that cells located closer to the innervating vascular branch exhibit greater resistance to stress, thereby benefiting from extended EMD processing durations. Further work is required to better understand whether CD34^dim^ subsets represent more mature or terminally differentiated populations and extrapolate the origins of both pericyte populations through the use of scRNAseq.

We found that, despite using a more standardized washing algorithm, we still experienced variability in cell subtype enrichment from EMD processing. Further analysis revealed that clogging, which occasionally occurs due to the presence of larger connective tissue parcels (<2 mm) within the LA sample, causes processing blockages that alter the true number of passes compared with the theoretical value. Additionally, since both raw and processed LA mix within the same reservoir, it is possible that reduced viscosity of emulsified fat [81] causes a gradient within the PD, preventing uniform mechanical processing. Furthermore, we looked at patient demographics and procedural data obtained from the patient consent process for any correlations that could explain these differences. One possibility could be cannula size [82], which varied between 4 and 5 mm. Another possibility is variation between anatomical location of adipose tissue harvest or history of obesity and massive weight loss, which have been shown to affect SVF composition [83,84]. We suspect that the varied macrophage enrichment observed reflects obesity-induced infiltration [85,86], with mechanical processing releasing otherwise inaccessible macrophage populations from within the periendothelium and interstitial spaces between adipocytes [87,88]. While differences in patient age and ethnicity were present, other published work has found that these factors do not affect the SVF composition [89,90]. These multiple sources of variance illustrate the difficulty in controlling variability when working with clinical adipose tissue samples and outline variables that must be controlled prior to elucidating any standardization benefits.

While our current PD offers sufficient washing efficiency and workflow benefits, there is room for improvement. The shape of the internal baffles was designed for both mixing and agitation of LA within the PD body. As seen in Figure 4, increasing the wash time greater than 1-min decreased the stem/progenitor cell populations. We speculate that this may have been caused by excessive agitation of tissue against the sharp angular baffles. During the phase-separation step (Figure 2C,D), we also observed that pockets of saline would become trapped under the baffle, creating a ledge effect. This would require an additional round of phase separation to fully rid the washed LA of saline. When removing washed LA from the system for final collection, these baffles would retain some LA tissue under the baffles, which resulted in ~10% loss. Future directions entail redesigning the shape and thickness of the baffles to prevent tissue loss and support good mixing without mechanically shearing the sample. Specifically, we will attempt to round the shape of the baffles, eliminating any edges, and reduce the height. We also plan to explore how reducing the length of baffling within the PD body affects washing efficiency and cell recovery.

A practical benefit that our new platform offers is reduced clogging compared to syringe pump processing, which is common while shuffling LA between two syringes and the EMD, particularly for larger cannulas (>4 mm). During processing, the syringe pump can stall despite being set at 100% force. It is possible that the distance between the PD chamber and the EMD (Figure 6A) allows for LA to reach the desired flow rate prior to entering the EMD’s 1.5 mm constriction. Alternatively, the tubing and Luer–Barb used to connect the EMD to the tubing flow path contain less distance for LA to travel through a 1.5 mm constriction compared to the use of a syringe. Another benefit of the PD is that it doubles as a reservoir for LA, which enables multi-device processing within the same platform. As a proof of concept, we integrated the EMD within the processing loop through the use of two three-way valves. The current setup produces ~50 mL of final LA product, which is reasonable for washing small-volume graft fillers and applications for SVF injections. However, we will explore scaling up the system in future work to process hundreds of mL per round, or perform multiple washings in parallel in separate PD devices.

## 5. Conclusions

In this work, we have presented a fluidic preparation device that can be utilized to wash human LA in a closed-loop system, enabling integrated downstream mechanical processing for fat grafting and adipose-based therapeutics. This platform produces adipose tissue of comparable quality to MF and maintains relative numbers of SVF stem/progenitor cells. Using the new iEMD processing protocol, our fluidic platform demonstrates enrichment of MSC, EPC, pericyte, and CD34^dim^ TA progenitor populations that is similar or superior to previous work using our EMD with a syringe pump and traditional nanofat processing. The closed-loop nature and use of peristaltic pumps enable automation and standardization of LA for clinical applications. Future work will seek to decrease preparation times through automation for both research and clinical applications and use in vitro and in vivo models to assess graft survival.

## 6. Patents

The subject matter of this manuscript is patent-pending and the intellectual property of the Regents of the University of California. 

## Figures and Tables

**Figure 1 bioengineering-12-00918-f001:**
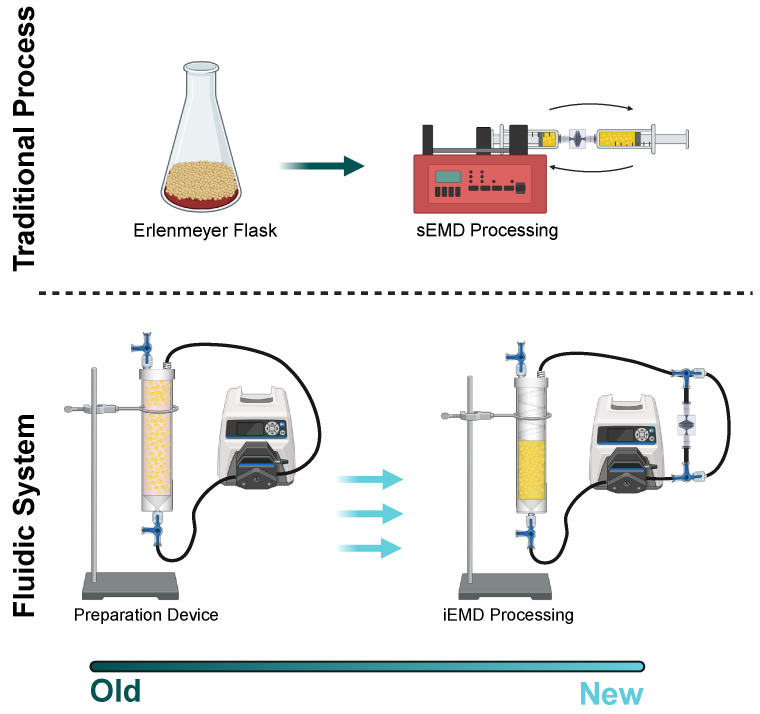
Platform for LA washing and mechanical processing. Traditionally, LA and saline are placed within a flask at a 1:1 ratio. The mixture is swirled around and allowed to phase-separate for 5 min. The infranatant layer is removed and this process is repeated ~3 times. Our new fluidic platform utilizes the PD with internal baffles to wash LA in a closed-loop system under peristaltic flow. An Emulsification and Micronization Device (EMD) can also be integrated via valving to perform mechanical processing within the same system and using the same peristaltic pump.

**Figure 2 bioengineering-12-00918-f002:**
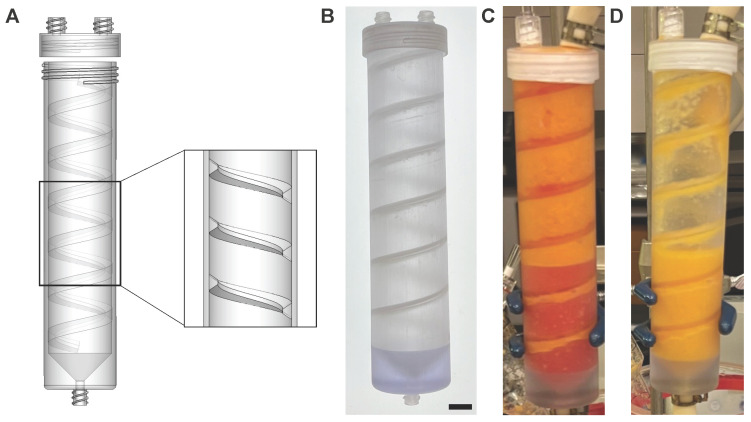
Preparation of device design and fabrication. (**A**) Schematic of PD showing internal baffles that are intended to enhance mixing as sample passes through the column. Expanded region in the cut-out shows the fin-like shape of the baffles. (**B**) Image of the device produced by three-dimensional printing. (**C**) Image of the PD during the first wash cycle after phase separation for 5 min. (**D**) Image of final LA after three rounds of washing and with the infranatant layer removed. Scale bar = 1 cm.

**Figure 3 bioengineering-12-00918-f003:**
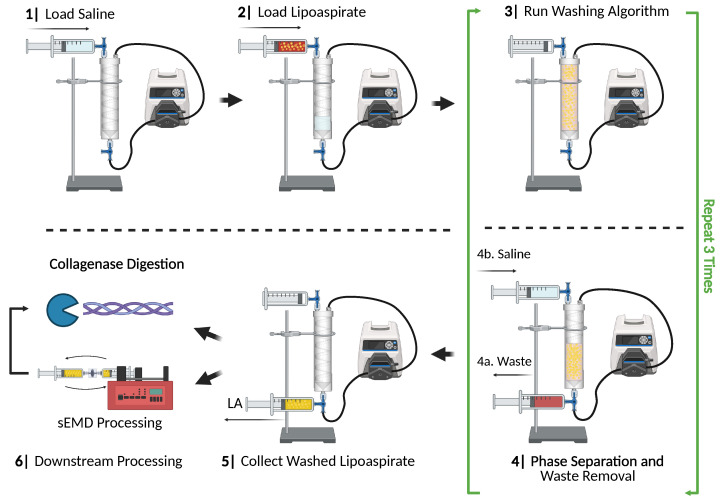
PD washing workflow. First, approximately 75 mL of saline is loaded into the PD through one of the inlets, followed by an equal volume of LA. After loading, PD and tubing contents are recirculated through the closed-loop system using a peristaltic pump. The sample in the PD is then phase-separated for 5 min, prior to removal of infranatant to waste. Fresh saline is then loaded into the PD and the wash algorithm is repeated. After three wash cycles, the washed LA is collected for downstream analysis.

**Figure 4 bioengineering-12-00918-f004:**
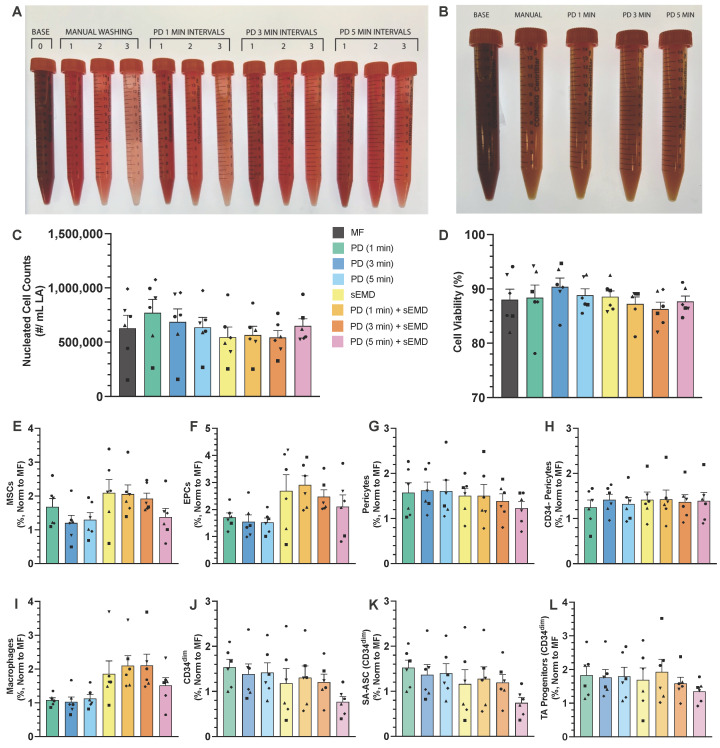
PD washing and evaluation of LA. Human LA (n = 6) was washed with the PD for 1-, 3-, or 5-min recirculation times. (**A**) Image of infranatant waste fractions collected during each wash cycle. Visually, the 1-min recirculation time appeared closest to manual washing as the 3- and 5-min times appeared slightly darker in color. (**B**) Image of LA after 3 wash cycles, compared to unwashed LA. No substantial difference in color difference was noted between samples. Washed LA was mechanically processed with the EMD using a syringe pump. All samples were digested with collagenase I to determine total cell count, viability, and evaluate cell subpopulations. (**C**) Nucleated cell counts decreased recirculation time slightly in the PD, as well as EMD processing, compared to MF. (**D**) Nucleated cell viability remained >85% for all conditions. Flow cytometry was used to quantify SVF subpopulations, which were normalized to MF (value = 1). Most cell populations, including (**E**) MSCs, (**F**) EPCs, (**G**) pericytes, (**I**) macrophages, (**J**) CD34^dim^ cells, (**K**) SA-ASC, and (**L**) TA progenitors, exhibited a slight downward trend in enrichment following EMD processing as recirculation time increased. In contrast, (**H**) CD34- pericytes remained stable across all recirculation and EMD processing conditions. Shapes denote individual patient replicates within this figure; assignments are not preserved across figures. Error bars represent standard error from at least 3 independent experiments.

**Figure 5 bioengineering-12-00918-f005:**
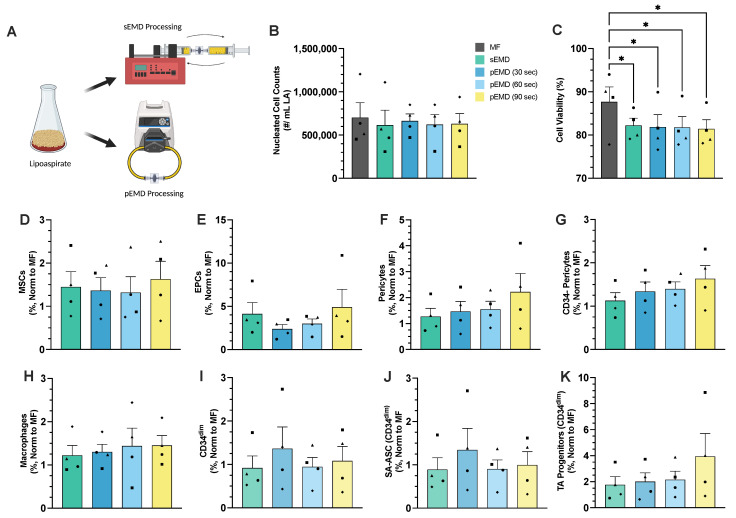
Peristaltic EMD processing. Human LA (n = 4) was manually washed and mechanically processed through the EMD using both syringe pump (sEMD) and peristaltic pump (pEMD) formats at 20 mL/s. sEMD utilized 30 passes, while pEMD recirculated for different time points. (**A**) Schematic demonstrating different formats, including tubing loop used for pEMD. (**B**) Nucleated cell counts were similar for MF and all EMD-processed conditions. (**C**) Viability was ~80% for all EMD conditions. Flow cytometry revealed a general but modest upward trend in enrichment for (**E**) EPCs, (**F**) pericytes, (**G**) CD34- pericytes, (**H**) macrophages, and (**K**) TA progenitor cells, with all conditions surpassing sEMD. (**D**) MSCs demonstrated a slight enrichment at 90 s, while (**E**) EPCs required at least 60 s of peristaltic processing to match sEMD processing. (**I**) The CD34^dim^ population and (**J**) SA-ASC saw greater enrichment at 30 s and 90 s compared to MF. Shapes denote individual patient replicates within this figure; assignments are not preserved across figures. Error bars represent standard error from at least 3 independent experiments. Graphs that violated assumptions for two-way ANOVA were analyzed using a non-parametric Friedman test with Dunn’s post hoc comparisons. Holm–Šidák post hoc; * *p* < 0.05.

**Figure 6 bioengineering-12-00918-f006:**
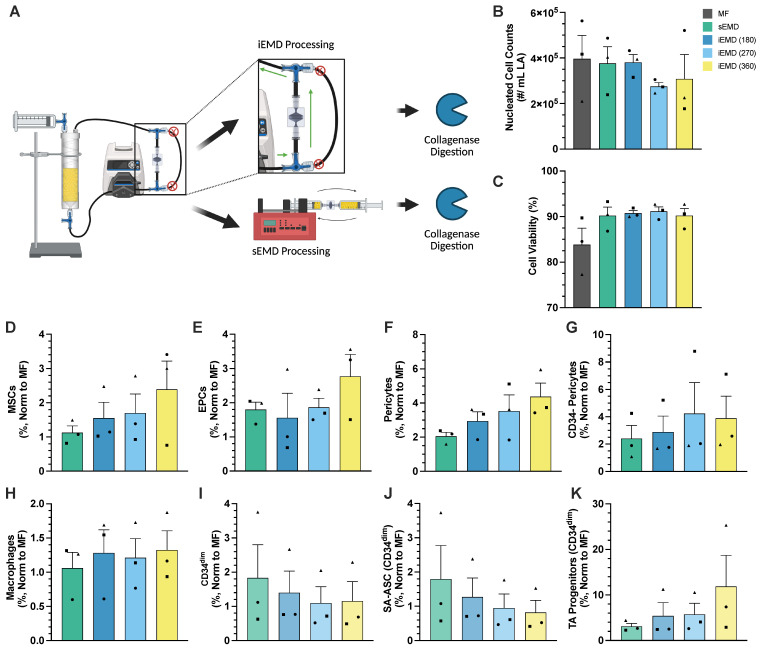
Fluidic platform enables washing and integrated mechanical processing. Human lipoaspirate (n = 3) was washed in a baffled PD system using a 1-min wash duration and was followed by either syringe EMD (sEMD) processing or integrated closed-loop EMD processing (iEMD). (**A**) Schematic detailing how the PD enables iEMD processing. Through the actuation of two three-way valves, a separate flow path is opened containing an EMD to allow for mechanical processing. The PD doubles as a reservoir during mechanical processing. (**B**) Nucleated cell counts slightly dropped among all iEMD processing conditions compared to sEMD processing. (**C**) Viability increased to ~90% upon EMD processing with either modality. Flow cytometry analysis revealed an upward trend in enrichment among (**D**) MSCs, (**E**) EPCs, (**F**) pericytes, (**G**) CD34- pericytes, (**H**) macrophages, and (**K**) TA progenitor cells, with (**E**) EPCs requiring 270 s of processing to match sEMD. (**I**) CD34^dim^ and (**J**) SA-ASC populations exhibited a downward trend in enrichment with iEMD vs. sEMD processing. Shapes denote individual patient replicates within this figure; assignments are not preserved across figures. Error bars represent standard error from at least 3 independent experiments.

**Table 1 bioengineering-12-00918-t001:** Stem and progenitor cell types of interest.

Cell Type	Marker	Significance
Macrophages	CD45^+^, CD34^−^, CD11b^+^	Regulate inflammation and tissue repair
Endothelial Progenitor Cells (EPCs)	CD45^−^, CD31^+^, CD34^+^	Promote vascularization of healing tissues; localized to luminal side
Mesenchymal Stem Cells (MSCs)	CD45^−^, CD31^−^, CD34^+^	Central to regenerative wound healing
Pericytes (Strategy 1)	CD45^−^, CD31^−^, CD146^+^	Support angiogenesis and maintain tissue homeostasis; localized to adventitia
Pericytes (Strategy 2)	CD45^−^, CD31^−^, CD146^+^, CD34^−^	(Same as above)
Transit-Amplifying (TA) Progenitor Cells	CD45^−^, CD31^−^, CD146^+^, CD34^+^	May give rise to EPC and SA-ASC populations
Supra-Adventitial Adipose Stromal Cells (SA-ASCs)	CD45^−^, CD31^−^, CD146^−^, CD34^+^	Reside around arterioles and venules

**Table 2 bioengineering-12-00918-t002:** Flow cytometry probe panel.

Assay	Probe
CD31	Anti-CD31 Ab (Clone WM59)–PE
CD34	Anti-CD34 Ab (Clone 561)–BV421
CD45	Anti-CD45 Ab (Clone 2D1)–BV510
CD11b	Anti-CD11b Ab (Clone M1/70)–FITC
CD146	Anti-CD146 Ab (Clone SHM-57)–APC
Viability	7-AAD

PE, phycoerythrin; FITC, fluorescein isothiocyanate; APC, allophycocyanin; 7-AAD, 7-aminoactinomycin D.

## Data Availability

The data presented in this study are available upon request from the corresponding author.

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
