# Peer review of "Integrated Fluidic Platform for Washing and Mechanical Processing of Lipoaspirate for Downstream Fat Grafting and Regenerative Applications"

_bioengineering, 2025, doi:10.3390/bioengineering12090918_

Round 1
Reviewer 1 Report
Comments and Suggestions for Authors
The manuscript needs revision. Please refer to comments given in the text of reviewed attached file of the manuscript.

Author Response
See file attached

Reviewer 2 Report
Comments and Suggestions for Authors
In this manuscript, “Integrated Fluidic Platform for Washing and Mechanical Processing of Lipoaspirate for Downstream Fat Grafting and Regenerative Applications” by Zalazar et al. reports a preparation device that is designed for peristaltic pump-driven washing of LA and can be seamlessly combined with our previously developed emulsification and micronization device into a closed-loop, automated platform. This work is well written and could be publish in the current form.
Author Response
No reviewer comments were provided. Response was publish as is